# Development of a Process for Domestic Wastewater Treatment Using *Moringa oleifera* for Pathogens and Antibiotic-Resistant Bacteria Inhibition under Tropical Conditions

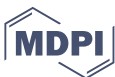

Nini Sané [1,2], Malick Mbengue [1], Amandine Laffite [3], Serge Stoll [3], John Poté [3] and Philippe Le Coustumer [2,4,*]

1  Laboratoire de Microbiologie Appliquée et de Génie Industriel, Ecole Supérieure Polytechnique, Université Cheikh Anta Diop, Dakar-Fann, Dakar 5085, Senegal; ninisane@gmail.com (N.S.); mbenguemalick@hotmail.com (M.M.)
2  Université de Bordeaux, UF Sciences de la Terre, Allée G. Saint Hilaire, 33615 Pessac, France
3  F.-A. Forel Department and Institute of Environmental Sciences, Faculty of Science, University of Geneva, 66 Boulevard Carl-Vogt, 1205 Geneva, Switzerland; amandine.laffite@gmail.com (A.L.); serge.stoll@unige.ch (S.S.); john.pote@unige.ch (J.P.)
4  Bordeaux Imaging Center UAR 3420 CNRS/UB–US 004 INSERM Centre Génomique Fonctionnelle Bordeaux 1er étage–Case 52, 146 rue Léo Saignat, CS 61292, 33076 Bordeaux, France
*  Correspondence: philippe.le-coustumer@u-bordeaux.fr; Tel.: +33-6-20-31-04-31

**Abstract:** Developing countries are confronted with general issues of municipal wastewater management and treatment. Untreated wastewater and faecal sludge from septic tanks and traditional toilets are often discharged into rivers and used for urban agriculture without any treatment to minimize potential biorisks. Such practices result in potential environmental and public health risks. In this study, a wastewater treatment plant prototype coupled with *Moringa oleifera* seeds treatment was developed to evaluate their effectiveness for the reduction of faecal indicator bacteria and antibiotic-resistant bacteria in domestic wastewater. We demonstrated that that the proposed wastewater treatment plant prototype reduces bacteria by 99.34%. A high removal of the bacteria load was obtained after the addition of *Moringa oleifera* seeds into waters, with removal rates of 36.6–78.8% for *E. coli*, 28.3–84.6% for faecal coliform, 35.3–95.6% for Vibrio cholera and 32.1–92.4% for total flora. A similar effect of *Moringa oleifera* seeds was noted for the removal of antibiotic-resistant bacteria, extended-spectrum beta-lactamases and carbapenem-resistant Enterobacteriaceae, with a removal rate of up to 98% for *E. coli* and faecal coliform, 100% for Vibrio cholera and 91.96% for total flora. This study demonstrated the high removal efficiency pathogens and antibiotic-resistant bacteria from domestic wastewater using *Moringa oleifera* seeds.

**Keywords:** domestic wastewater; biological contamination; wastewater treatment plant; *Moringa oleifera*; antibiotic resistance

## 1. Introduction

Addressing the challenges of access to safe drinking water and sanitation remains a global concern, particularly in developing countries. The United Nations' sustainable development goals establish a program to achieve access to adequate sanitation, the eradication of open defecation, halving of the proportion of untreated wastewater, increases recycling and safe reuse globally [1]. Sectors such as agriculture are turning to wastewater reuse to overcome challenges related to water availability. However, wastewater reuse will likely lead to the contamination of crops, soil, water and resources and resistance to antibiotics [2]. Currently, immediate solutions are required regarding antibiotics resistance, which is considered the third largest threat to public health [3]. The excessive use of antibiotics for the treatment of human and animal diseases contributes to an increase in the antibiotic-resistant bacteria that are present in feces. These resistant bacteria are retransmitted into wastewaters

and contribute to the development of antibiotics resistance genes that can be dangerous in case of the reuse of waters and sludges for agricultural propose [4,5]. Faced with these concerns, several questions arise when it comes to strengthening wastewater treatment plants in sub-Saharan Africa and to optimizing the reuse of liquid and solid waste from individual sanitation facilities in the agricultural sector.

In wastewater treatment, coagulants are used for the reduction of turbidity, leading to the reduction of contaminants. The most widely used chemical coagulants are aluminum and iron sulphate. The use of chemical coagulants presents many concerns, as aluminum residues are reported to cause neurodegenerative diseases such as Alzheimer's [6]. These coagulants are reported to not be sustainable due to the high carbon footprint associated with their production, but also to the generation of excessive nonbiodegradable sludge [7–9]. Issues related to the use of chemical coagulants necessitate the use of plant-based coagulants such as *Moringa oleifera* that are cost-effective, biodegradable, eco-friendly and in line with the sustainable development goals [8,10]. *Moringa oleifera* seeds (MO$_S$), as well as other components of the moringa tree, are increasingly used in the water treatment process as naturel coagulants. Much literature on the application of Moringa oleifera seed extract in water treatment indicates that MO$_S$ used as a coagulant in wastewater treatment may be efficient enough to be considered as an alternative to chemical coagulant use [11]. The main advantage of using MOs as natural coagulants is that they do not have any notable drawbacks, and they are non-toxic and non-corrosive [12]. In addition, *Moringa Oleifera* seeds are easily accessible for most developing countries, and they are fully biodegradable. A number of studies reported the use of MO$_S$ to reduce faecal indicator bacteria (FIB) [13–15] and to improve wastewater treatment using filtration methods [16–18]. It was demonstrated that the glucomoringin present in MO$_S$ removes up to 99.99% of total coliform [18]. Other compounds of the Moringa oleifera tree such as leaves have also shown their effectiveness in reducing the antibiotics resistance acquired by bacteria [19].

In this study, we evaluated the removal efficiency of a wastewater treatment pilot (WWTP) prototype to eliminate faecal contamination and antibiotic-resistant bacteria from domestic wastewaters. According to the objective of treated wastewater reuse for agriculture purposes, national and internationals standards need to be reached. According to section III of the Senegalese Sanitation Regulations (art. L76), the qualitative characteristics of domestic wastewater reuse are set, and the quantity of *E. coli* after treatment must be $\leq 10$ CFU mL$^{-1}$ for restricted irrigation [20]. Without more directives concerning unrestricted irrigation, especially market gardening, we therefore based our study on French regulations. According to the French regulations [21], the concentration of *E. coli* in reused wastewater is set depending on the sanitary quality targeted. In fact, 4 levels of sanitary quality (A, B, C, D) of wastewater are defined in French regulations, and requirements vary for each level. After treatment by the bacterial filter, the quantity of *E. coli* in wastewater attained an average of 270 CFU mL$^{-1}$, which refers to a sanitary quality level $C \leq 1000$ CFU mL$^{-1}$. The level C allowed the reuse of treated wastewater for nurseries and shrubs and other floral crops, as well as cereal and fodder crops, using localized irrigation. It also allowed the cultivation of fruit trees only by drip. However, our objective remains to attain the level A that would allow for the use of water for market gardening, fruit and vegetables crops not transformed by an adapted industrial heat treatment (except cress culture) and the maintenance of green spaces open to the public. In this context, we propose herein to evaluate the effectiveness of an additional treatment using MO$_S$ in order to achieve better sanitary quality of reused wastewater.

The evaluation is based on the quantification of faecal indicator bacteria (FIB), including *E. coli*, *faecal coliform*, *faecal streptococcus*), total bacteria flora and *Vibrio cholera*. The sensibility of the bacteria to antibiotics is also tested by adding ampicillin on agar and using counting methods for extended-spectrum beta-lactamases (ESBL) and carbapenem-resistant Enterobacteriaceae (CRE). The sensibilities of FIB, *Vibrio cholera* and antibiotic-resistant bacteria to MO$_S$ were tested by adding seed powder into wastewater samples. It was found that several studies have been carried out previously to study the impact of Moringa oleifera

bacteria inhibition, but this could be the first study on MOs inhibiting antibiotic-resistant bacteria under "tropical conditions" in a sub-Saharan African country.

## 2. Materials and Methods

### 2.1. Conception of Wastewater Treatment Plant Prototype

A non-collective sanitation system treats wastewater without external energy input. An individual sanitation plant is made up of reactors that carry out the pretreatment, purification and discharge of domestic wastewater. The wastewater treatment plant prototype that we designed is composed of three tanks. The entry tank (Figure 1) works like a conventional septic tank, in which an anaerobic biological treatment consumes/mineralizes organic matters. The second tank (bacterial filter) provides treatment by filtration and is equipped with an air pipe to ensure aerobic treatment. A second filtration step is carried out in the well before water discharge into soil. The sizing was carried out according to the AFNOR regulations of 1983 and 1998 transcribed in the non-collective sanitation guide of 2016 [22]. The sizing was achieved according to the number of users in houses, and thus the plant size is correlated with house size. Our prototype was designed to accommodate for a standard house that accommodates 12 permanent users, which corresponds to a house with five main rooms.

After conception and sizing, we realized a septic tank with a useful volume equal to 3 $m^3$, a bacterial filtration that contains 1.6 $m^3$ of filtering porous material composed by basalt with diameters between 3 to 40 mm and an infiltration well offering a contact area equal to 10 $m^2$. The WWTP prototype was built in 2017 in Keur Moussa, located about fifty kilometers east of Dakar (Appendix A).

Wastewater is transferred from the house using a 110 mm PVC pipe. The entrance of the septic tank is equipped with a 50 cm-high deflection wall (Figure 1a) to redirect the water downwards. Another deflection wall was placed before the exit of the septic tank to regulate the outgoing flow. The inlet pipe, if 15 cm higher than the outlet pipe, prevents water from backing up to the house. A WWTP prototype does not work in continuous flow but rather in overflow. When the water depth inside the septic tank reaches 1.5 m, the supernatant was released into the bacterial filter. The flow rate as well as the hydraulic retention time vary according to the daily quantity of water used in the house.

Water from the septic tank is transferred into the bacterial filter through three perforated pipes (Figure 1b). The pre-treated effluents are distributed on the bed of filtering materials made of basalt. There is filtration through the layers, and filtered waters are transferred into the infiltration well by three perforated pipes. A second filtration through the layers of basalt (Figure 1c) is carried out, after which waters are discharged into the natural ground composed of sand.

### 2.2. Sampling Procedure

Wastewaters from the WWTP prototype were collected aseptically after homogenization in November 2020 with temperatures around 29 $\pm$ 1 °C. Wastewaters were collected at three points: the entrance of the septic tank (F0), the entrance of the bacterial filter (F1) and entrance of the infiltration well (F2), and then filled into 250 mL previously cleaned plastic bottles. After sampling, samples were stored in an icebox at 4 °C and transported to the Department F.-A. Forel of the University of Geneva for analysis within 48 h.

### 2.3. Sample Treatment with Moringa oleifera Seeds

The seeds of M. oleifera were purchased from a plantation in Gaaya, Saint Louis, Senegal. Seeds must be completely dry on the tree before picking. Pods should be brown in color because green pods do not have active coagulation agents. Dry *Moringa* seeds were first removed from their teguments and husk. White kernels were ground to a fine powder using a pestle and a mortar and sieved through a 1 mm sieve. The MO powder was then added to wastewater. Approximately 250 mL of wastewater F0, F1 and F2 were prepared in glass bottles. MO powder was added in each assay to a final concentration

of 300 mg $L^{-1}$ for samples collected from the septic tank and 50 mg $L^{-1}$ for samples collected from the bacterial filter and the infiltration well. The incubation was set for three times: rapid stirring for 90 s to destabilize the colloids, slow stirring for 5 min to forms micro-flocs and decantation of about 3 h on a multi-position magnetic stirrer. All samples were then centrifuged at 4000 rpm for 10 min. The supernatant was then decanted for microbiological analysis.

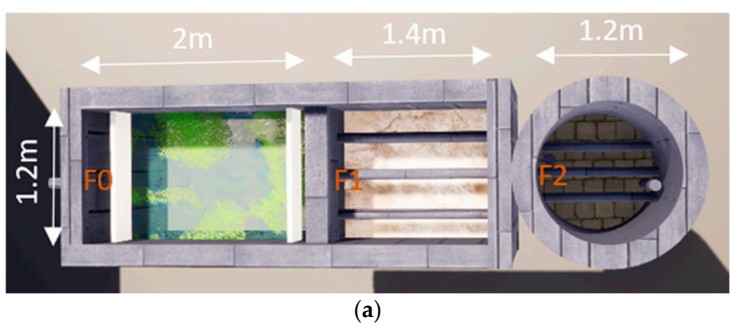

(**a**)

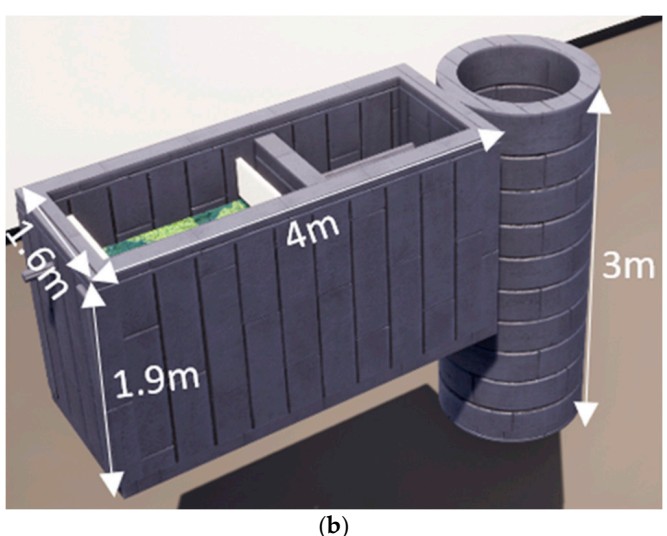

(**b**)

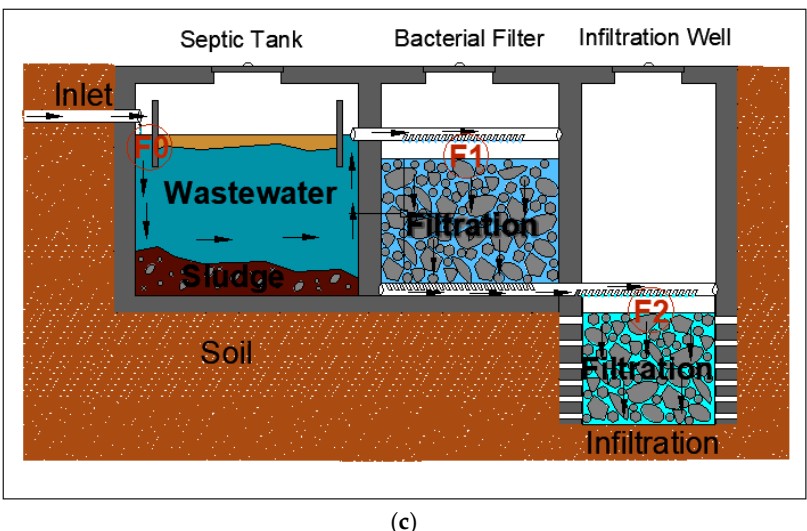

(**c**)

**Figure 1.** Three-dimensional dimensions of the wastewater treatment plant (WWTP) prototype. (**a**) Top view of the WWTP prototype; (**b**) longitudinal view of the WWTP prototype; (**c**) operating outline of the WWTP prototype.

In order to identify the impact of MOs on antibiotic resistant bacteria, a bacterial culture was carried out for samples prepared with MOs in the presence of ampicillin, which is an antibiotic belonging to beta-lactam. According to the CLSI breakpoints (Clinical and Laboratory Standards Institute) M100 sheet [17], the following doses of antibiotics were added to the culture media prepared for the identification of *E. coli*, faecal coliforms, enterococcus, vibrio cholera and total flora:

– 32 mg/L for enterobacteria, in particular faecal coliforms and *E. coli*
– 16 mg/L for enterococcus, vibrio cholera and total flora.

In addition, a bacterial culture of samples with MO was carried out on medium ESBL for extended-spectra-β-lactam resistance and CRE for carbapenem resistance.

### 2.4. Microbial Analysis of Water Samples

The microbial analysis was performed to evaluate the effectiveness of the WWTP prototype for bacterial removal and the impact of the *Moringa oleifera* seeds' addition to the process. Five faecal pollution markers (*Escherichia coli*, faecal coliforms, *Enterococcus* spp., *Vibrio cholera* and total heterotrophic bacteria) were quantified according to international regulations that recommend the use of these indicators to assess the efficiency of wastewater treatment, the quality of drinking water and the hygienic safety of recreational waters [23–26]. *Escherichia coli*, faecal coliforms, *Enterococcus* spp., *Vibrio cholera* and total heterotrophic bacteria were quantified using the plate pouring method as described by Rodier et al. [27]. Bacteria were dispersed in a solid media and incubated under favorable conditions for the growth of isolated colonies that could be counted. Microbiological selective media included tryptone bile x glucuronide (TBX) for the quantification of *Escherichia coli*, violet-red bile lactose agar (VRBL) for the quantification of faecal coliform, Bile Esculin sodium Azide (BEA) for the quantification of *Enterococcus* spp., Reasoner's 2A Agar (R2A) for the estimation of the total heterotrophic flora and thiosulfate-citrate-bile-saccharose (TCBS) for the quantification of *Vibrio* spp. The first step of the microbiological analysis was the preparation of agar media according to manufacturer's recommendations. After that, we proceeded to sample dilution, which was performed with sterile saline water (0.9% NaCl). Five mL of wastewater was added into a falcon tube containing 45 mL of saline solution and shaken to obtain the first dilution. To obtain the second dilution, 5 mL from the first one was added to 45 mL of saline solution and shaken. The same procedure was used until n dilutions ($n$ = 6 in this study). One mL of each diluted sample was placed in the center of a sterile Petri dish plate. Molten cooled selective agar was then poured into the Petri dish and mixed well. The reproducibility of the experimental procedure was estimated by means of triplicates on samples. After solidification, plates were inverted and incubated. The temperature and time of incubation for each medium are described in Table 1. After incubation, colonies were counted, and results were reported as UFC mL$^{-1}$ considering the dilution factor. The number (N) of colonies was calculated with the following equation for two successive dilutions, and only plates with more than 15 or less than 300 colonies were taken into account:

$$N = \frac{\sum C}{1.1\,D};$$

– C = average of colonies counted for each dilution
– D = dilution rate of the first dilution considered

Antibiotic-resistant bacteria (antibiotic-resistant *E. coli*, antibiotic-resistant faecal coliform, antibiotic-resistant *Enterococcus* spp., antibiotic-resistant *Vibrio cholera*, antibiotic-resistant total heterotrophic flora) were quantified by supplementing selective media with ampicillin (32 mg/L for Enterobacteriacae and total heterotrophic flora; 16 mg/L for *Enterococcus* spp.) according to the CLSI breakpoints (Clinical and Laboratory Standards Institute) M100 sheet [28]. Extended-spectra-β-lactam and carbapenem-resistant

bacteria were isolated using Chromagar$^{TM}$ ESBL and mSuperCARBA media (Chromagar, Paris, France).

**Table 1.** Characteristics of nutritive agars used.

| Marker | Nutritive Agar | Brand | Reference | LOT | Incubation |
|---|---|---|---|---|---|
| *E. coli* | TBX | Biolife Italiana | 402 1562 | ML 5507 | 18 to 24 h at 44 °C |
| Faecal coliform | VRBL | Biolife Italiana | 402 1852 | LD7103 | 18 to 24 h at 44 °C |
| Faecal streptococcus | BEA | Biolife Italiana | 401 0182 | FP5303 | 24 h at 44 °C |
| *Vibrio cholera* | TCBS | Oxoid Ltd. | 190 0511 | CM0333 | 24 h at 37 °C |
| Total flora | R2A | Biolife Italiana | 401 9962 | MC2801 | 5 days at room T° |

Note(s): TBX: tryptone bile x glucuronide; VRBL: violet-red bile lactose agar; BEA: bile esculin sodium azide; TCBS: thiosulfate citrate bile saccharose; R2A: reasoner's 2A agar.

*2.5. Data Analysis*

Data analysis was performed using Excel software. The WWTP removal efficiency was estimated considering the abatement rate between the entrance and the exit of the wastewater treatment plant in order to evaluate the performance of the system. The impact of MO on pollution removal was estimated by abatement rate with and without additional *Moringa* treatment. Statistical analysis was performed using R software (version 4.0.4) and the Rcmdr package after log10 reduction. Statistical significance was defined by 95% confidence intervals ($p < 0.05$).

**3. Results and Discussion**

*3.1. Abatement of Bacteria in WWTP Prototype*

The purification potential of the WWTP prototype was determined by quantifying five bacterial markers (*E. coli*, faecal coliform, *Enterococcus* spp., *V. cholera* and total heterotrophic flora) in water samples before and at each step of the treatment process (Table 2). For example, at the entrance of the septic tank (F0), the average concentration of bacterial markers was $1.97 \times 10^4 \pm 2.52 \times 10^3$, $2.13 \times 10^4 \pm 1.59 \times 10^4$, $3.30 \times 10^3 \pm 4.58 \times 10^2$, $2.63 \times 10^7 \pm 4.35 \times 10^6$ CFU mL$^{-1}$ for *E. coli*, faecal coliform, *V. cholera* and total heterotrophic flora (Table 2). However, the abundance of *Enterococcus* spp. bacteria was under the limit of quantification for all samples. After decantation (F1), *E. coli* and total heterotrophic flora abundances increased by 2.5 and 6 times, respectively, and faecal coliform and *V. cholera* abundances decreased by 27.2% and 59.0%, respectively. At a global level, based on the observation between F0 and F1 concerning the increase in total flora load and *E. coli*, the entering tank could be assimilated as a bacteria concentrator due to the environment (temperature between 28.9 °C and 30.10 °C; presence of organic matters), which is favorable for the proliferation of bacteria. Other studies have highlighted the growth of *E. coli* inside the septic tank as mentioned by Appling et al., who reported an increase in *E. coli* by 100-fold into septic tanks under Georgian summer weather with temperatures around 30 °C [29]. The filtration step allowed for a reduction in bacterial abundances from 98.1 to 99.4% between the entrance of the bacterial filter (F1) and the entrance of the infiltration well (F2). An ANOVA test showed a significant difference (ANOVA 2, $p < 0.01$) in bacterial abundance after the combined effect of decantation (in the tank) and filtration (in the bacterial filter), which allowed a bacterial purification from 89.9–98.6% for FIB and up to 99.3% for *Vibrio cholera*. The number of obtained *Vibrio cholera* strains was low in comparison with the counting number obtained for FIB strains; this could explain the high

abatement rate obtained for *Vibrio cholera*. Consequently, the experimentation with more strains of *Vibrio cholera* is recommended.

**Table 2.** Abundance of bacteria in wastewater.

| Samples | *E. coli* | | Faecal Coliform | | *Vibrio cholera* | | Total Flora | |
|---|---|---|---|---|---|---|---|---|
| | CFU mL$^{-1}$ | SD | CFU mL$^{-1}$ | SD | CFU mL$^{-1}$ | SD | CFU mL$^{-1}$ | SD |
| F0 | $1.97 \times 10^4$ | $2.52 \times 10^3$ | $2.13 \times 10^4$ | $1.59 \times 10^4$ | $3.30 \times 10^3$ | $4.58 \times 10^2$ | $2.63 \times 10^7$ | $4.35 \times 10^6$ |
| F1 | $5.00 \times 10^4$ | $1.00 \times 10^4$ | $1.55 \times 10^4$ | $1.17 \times 10^3$ | $1.35 \times 10^3$ | $3.49 \times 10^2$ | $1.58 \times 10^8$ | $4.05 \times 10^7$ |
| F2 | $2.73 \times 10^2$ | $3.79 \times 10^1$ | $2.97 \times 10^2$ | $2.52 \times 10^1$ | $2.17 \times 10^1$ | 4.93 | $2.65 \times 10^6$ | $1.37 \times 10^5$ |
| F0$_{MOS}$ | $5.47 \times 10^3$ | $2.08 \times 10^2$ | $3.77 \times 10^3$ | $6.66 \times 10^2$ | $1.12 \times 10^3$ | $5.36 \times 10^2$ | $7.00 \times 10^6$ | $1.48 \times 10^6$ |
| F1$_{MOS}$ | $1.06 \times 10^4$ | $7.21 \times 10^2$ | $1.11 \times 10^4$ | $2.54 \times 10^3$ | $5.90 \times 10^1$ | $2.21 \times 10^1$ | $1.20 \times 10^7$ | $3.17 \times 10^6$ |
| F2$_{MOS}$ | $1.73 \times 10^2$ | $3.51 \times 10^1$ | $4.57 \times 10^1$ | $9.02 \times 10^0$ | $1.40 \times 10^1$ | 6.24 | $1.80 \times 10^6$ | $3.46 \times 10^5$ |

### 3.2. Impact of Moringa oleifera Seeds on Bacteria Abatement

A laboratory experiment was conducted to determine the impact of MO$_S$ as natural coagulants during the wastewater treatment process. The results of the bacterial abundance in wastewaters (F0$_{MOS}$, F1$_{MOS}$, F2$_{MOS}$) are presented in Table 2. The abatements obtained after MO'$_S$ addition into wastewaters vary significantly according to the samples and type of bacteria (ANOVA, $p < 0.01$) (Figure 2). The abatements varied from 36.6% to 78.8% for *E. coli*, from 28.3% to 84.6% for faecal coliform, from 35.3% to 95.6% for Vibrio cholera, and from 32.1% to 92.4% for total heterotrophic flora. The highest abatement rates were obtained in samples F1 (at the entrance of the bacterial filter) for *E. coli*, Vibrio cholera and total flora, whereas for faecal coliform it was obtained in F2. From a global perspective, the quantity of bacteria at the exit of the septic tank (F1) was higher than the entrance (F0), with an increase of 83%, which is certainly due to the high bacterial activity during mineralization. This increase in the bacterial load could explain the more accentuated effect of MO on F1 waters. However, even if the effect of MO is more visible in F1, it is noted that the reduction in F0 is as important for all markers, with a reduction of 72.2% for *E. coli*, 82.34% for faecal coliforms, 65.96% for Vibrio cholera and 73.42% for total flora. Some studies have shown that the higher the turbidity of the wastewater, the more effective the reaction of the MO (without oil extraction) is [30]. Due to coagulation-flocculation-decantation actions, the reduction in the bacterial load being correlated to the reduction in the turbidity could explain the above results. Indeed, previous studies we have caried out on the WWTP prototype have enabled us to determine the average turbidity of the water, which is 97 NTU for F0, 73 NTU for F1 and 17.39 NTU for F2. However, additional studies should be carried out to for the particular case of faecal coliforms to confirm or invalidate the results enumerated above.

Our results are similar to those obtained by Vunain et al. [13] with a reduction of 96.6% in bacteria load, 60.5% for coliform and 97.3% for *E. coli* after using a concentration of 15 g L$^{-1}$ of MO$_S$ and 2 hours of contact time. A lower abatement rate (47%) for *E. coli* reduction was obtained by Poumaye et al. [16] after using a MO solution stirred into distilled water before being added to the wastewater, and Delelegn et al. [31] obtained a greater abatement rate (97%) using the same method with a higher concentration of Moringa seed powder. Considering the organic matter contained in MO$_S$ that can promote bacterial growth, few studies recommend completing the MO treatment with a filtration step using a sand filter [16,17].

According to the abatement rates obtained with the MO'$_S$ addition, in Figure 3a, a projection is proposed that gives an overview of the effect that MO would have if added on-site directly at the entrance of the septic tank (F0), at the entrance of the bacterial filter (F1) or the entrance of the infiltration well (F2). The projection was made on the basis of the reductions obtained between F0-F1F1-F2 and F0-F2 and the reductions obtained after the addition of MO into samples. The best combination for *E. coli* reduction consists of adding MO in the septic tank. Adding MO at the entrance of the septic tank would reduce the quantity

of *E. coli* from $1.97 \times 10^4 \pm 2.52 \times 10^3$ CFU mL$^{-1}$ to $5.47 \times 10^3 \pm 2.08 \times 10^2$ CFU mL$^{-1}$. Based on the increase obtained between F0 and F1 (2.5 times) and the reduction effect between F1 and F2 ($-99.4\%$), we could end up with a quantity of *E. coli* of 49 CFU mL$^{-1}$. *E. coli* abundance that would therefore refer to the level B ($\leq 100$ CFU mL$^{-1}$) defined by French regulation for wastewater reuse. Level B allows the use of treated wastewater (with some restrictions) for agricultural proposes such as market gardening, fruit and vegetable crops transformed by suitable industrial heat treatment, floral crops, cereal and fodder crops, pasture, etc. However, concerning the other markers studied, the best combinations for bacteria reduction obtained after projection remains adding MO in F2 for faecal coliform, in F1 for *Vibrio cholera* and in F1 for total flora. Considering these observations that can vary depending on markers, reactors and their characteristics, additional analyses after the addition of MO on-site would therefore be necessary in order to determine the best combination of treatment (using our WWTP prototype and MO$_S$) to adopt. On-site and large-scale experimentation would allow for setting the final process using MO as a natural coagulant under specific temperature and pH conditions. Indeed, the efficiency of MO can be influenced by water temperature, which acts on the coagulation process and floc formation. It was reported by Fitzpatrick et al. that warmer temperatures produce bigger flocs that settle quickly [32]. Since our study was conducted under tropical conditions, and the temperature into the WWTP prototype varies from 27 °C to 31 °C, the addition of MO onsite should be just as effective.

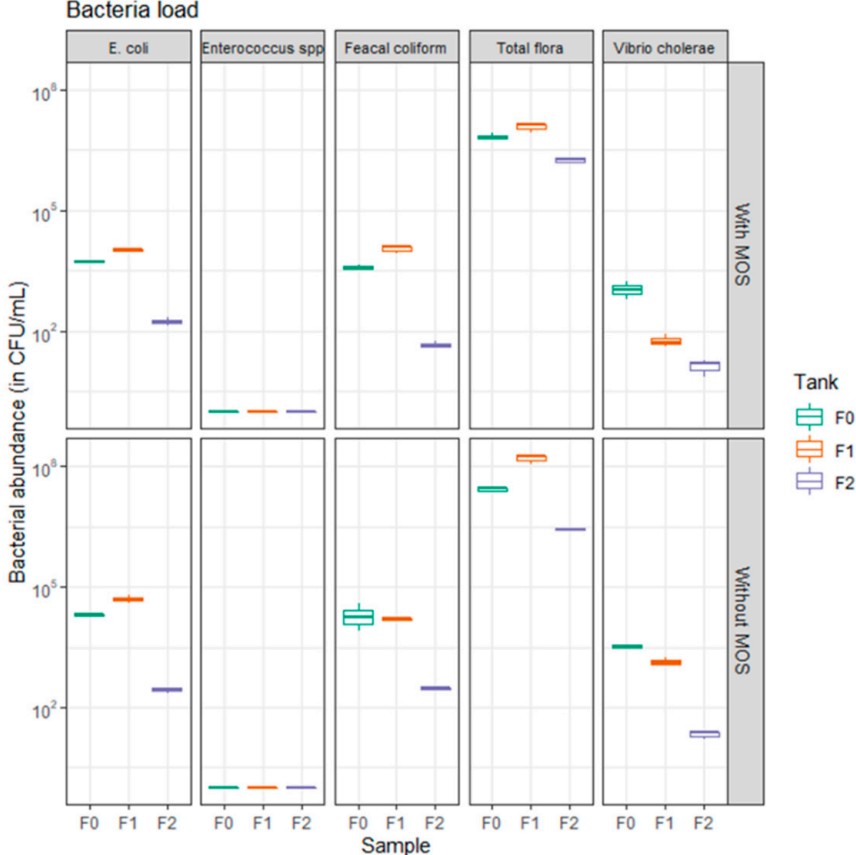

**Figure 2.** Abundances of bacteria in samples taken at the entrance of each tank for evaluation of abatement; counted number of bacteria represented as log 10 reduction.

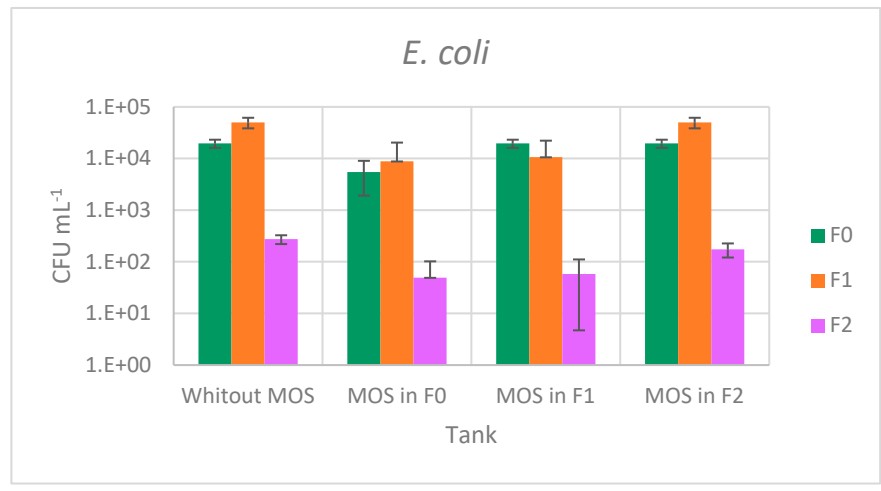

(**a**)

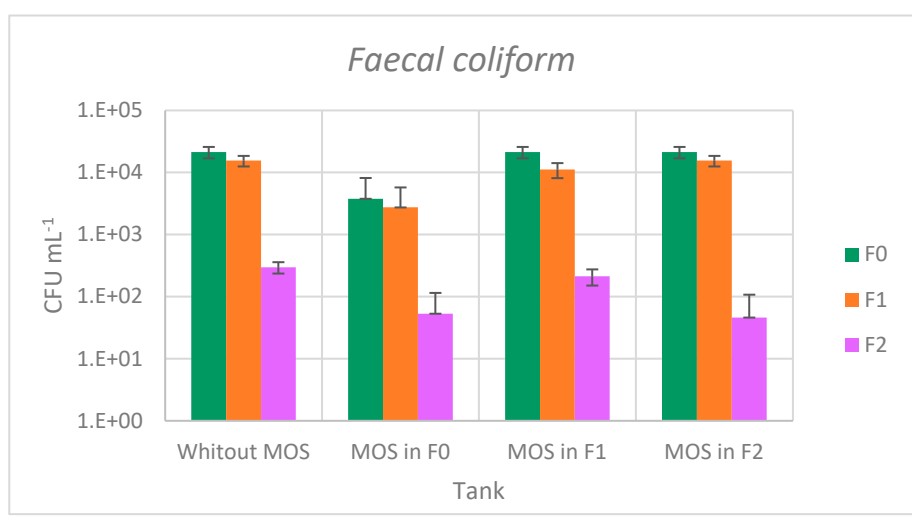

(**b**)

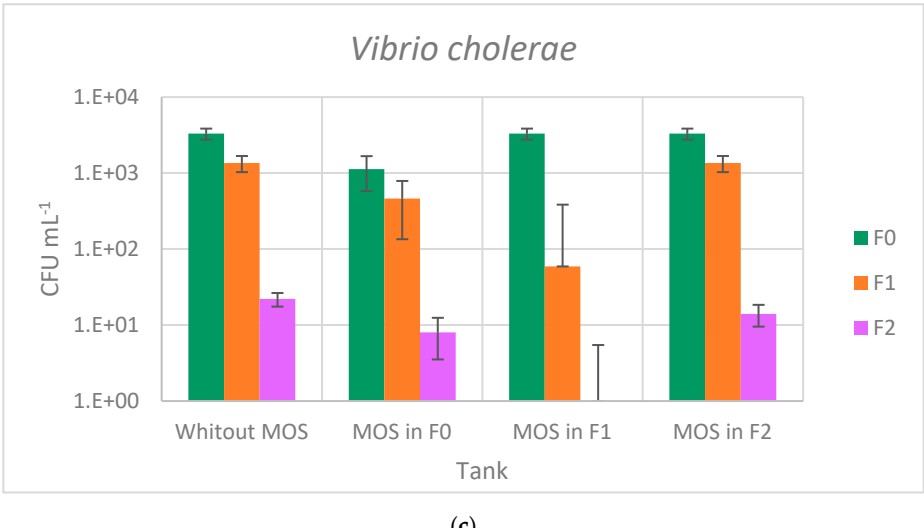

(**c**)

**Figure 3.** *Cont.*

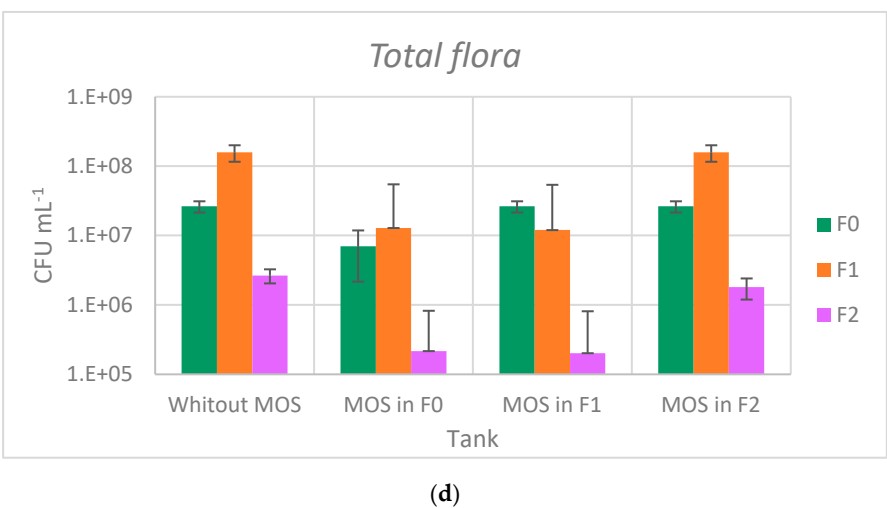

(**d**)

**Figure 3.** Projection of MO's effect on the entire treatment plant prototype after MO's addition in F0, F1, F2; (**a**): abundance of *E. coli*; (**b**): abundance of faecal coliform; (**c**): abundance of Vibrio cholera; (**d**): abundance of total flora.

### 3.3. Prevalence and Abatement of Antibiotic-Resistant Bacteria in WWTP Prototype

The prevalence of antibiotic-resistant bacteria (ARB) was assessed in our WWTP prototype. The results are presented in Table 3. The abundance of antibiotic resistant bacteria varies from $4.30 \times 10^2 \pm 1.23 \times 10^2$ to $1.92 \times 10^6 \pm 1.18 \times 10^5$ CFU mL$^{-1}$ in F0, from $1.47 \times 10^1 \pm 9.02$ to $1.80 \times 10^7 \pm 2 \times 10^5$ CFU mL$^{-1}$ in F1 and from $4.67 \pm 4.04$ to $6.83 \times 10^5 \pm 1.21 \times 10^5$ CFU mL$^{-1}$ in F2. At the entrance of the septic tank (F0), resistant isolates represented 75.9%, 65.9%, 13%, 7.3% of *E. coli*, faecal coliform, *Vibrio cholera* and total heterotrophic flora. respectively. At the entrance of the bacterial filter (F1), resistant isolates represented 25.1%, 83.7%, 1.1%, 11.4% of *E. coli*, faecal coliform, *Vibrio cholera* and total flora, respectively. At the entrance of the infiltration well (F2) resistant isolates represented 75.39%, 57.3%, 21.5%, 25.8% of *E. coli*, faecal coliform, *Vibrio cholera* and total flora, respectively. Summerlin et al. reported 81% of resistant isolates from *E. coli* among a total of 140 isolates, and this high percentage is in line with the results of this study [33]. From a global level, the ratio of antibiotic-resistant bacteria represented 7.38% of the total load in F0, 11.43% in F1 and 25.79% in F2. An increase in the percentage of resistant bacteria is observed from F0 to F2, therefore highlighting antibiotic-resistant genes' (ARG) proliferation in the tank microbiome. This increase should be correlated with the variation in temperature, which is higher in the infiltration well. Pasmionka et al. reported that the number of isolated antibiotic-resistant *E. coli* were higher in summer than winter [5]. Mao et al. [34] analyzed the abundance of 23 ARGs along the WWTP's process and highlighted an ARG enrichments ratio from 8 to 268. Summerlin et al. noticed an increase in ciprofloxacin-resistant strains' prevalence throughout the treatment process. Wastewaters are reported in other studies to be a source of antibiotic resistance and to contribute to the prevalence of anti-microbial resistance, antibiotic-resistant bacteria and antibiotic-resistant genes [35,36]. Despite the antibiotic-resistant enrichment throughout the treatment process, our WWTP prototype helped to significantly decrease antibiotic-resistant bacteria abundance by 98.6% for *E. coli*, 98.8% for faecal coliform, 98.9% for *Vibrio cholera* and 64.35% for total heterotrophic flora (ANOVA, $p < 0.01$).

Antibiotic resistance was further investigated for ESBL (Extending-Spectra-β-Lactamase) and carbapenem resistance in Enterobacteriaceae. In F0, the abundance of the *E. coli* and KESC group (e.g., *Klebsiella*, *Enterobacter*, *Serratia* and *Citrobacter*) resistant to beta-lactam were $279 \pm 47$ and $3851 \pm 315$ CFU mL$^{-1}$, respectively, whereas the abundance of *E. coli* and *KESC* resistant to carbapenem were $13 \pm 8$ and $1714 \pm 157$ CFU mL$^{-1}$. Resistance to antibiotics was lower in F1, with an abundance of $14 \pm 0$ and $101 \pm 28$ CFU mL$^{-1}$ for *E. coli* and *KESC* resistant to beta-lactam, and the abundance of *E. coli* and *KESC* resistant to car-

bapenem were $1 \pm 0$ and $384 \pm 29$ CFU mL$^{-1}$. No resistance to beta-lactam or carbapenem were detected in F2. An Extending-Spectra-β-Lactamase Escherichia Coli (ESBLEC) prevalence of 1.4% was 10 times higher than data published by Bréchet et al. [37]. However, our WWTP prototype was designed for 12 inhabitants, and ESBLEC prevalence in the WWTP prototype cannot be representative of the prevalence of ESBLEC in the population.

**Table 3.** Abundance of antibiotic-resistant bacteria in wastewater.

| Samples | AREC | | ARFC | | ARVC | | ARTF | |
|---|---|---|---|---|---|---|---|---|
| | CFU mL$^{-1}$ | SD | CFU mL$^{-1}$ | SD | CFU mL$^{-1}$ | SD | CFU mL$^{-1}$ | SD |
| F0 | $1.49 \times 10^4$ | $2.16 \times 10^3$ | $1.41 \times 10^4$ | $2.32 \times 10^3$ | $4.30 \times 10^2$ | $1.23 \times 10^2$ | $1.92 \times 10^6$ | $1.18 \times 10^5$ |
| F1 | $1.25 \times 10^4$ | $5.03 \times 10^2$ | $1.30 \times 10^4$ | $8.89 \times 10^3$ | $1.47 \times 10^1$ | 9.02 | $1.80 \times 10^7$ | $2.00 \times 10^5$ |
| F2 | $2.03 \times 10^2$ | $7.51 \times 10^1$ | $1.70 \times 10^2$ | $2.65 \times 10^1$ | 4.67 | 4.04 | $6.83 \times 10^5$ | $1.21 \times 10^5$ |
| F0$_{MOS}$ | $1.23 \times 10^3$ | $5.77 \times 10^1$ | $1.10 \times 10^3$ | $1.00 \times 10^2$ | $2.20 \times 10^2$ | $1.11 \times 10^2$ | $1.34 \times 10^6$ | $7.21 \times 10^4$ |
| F1$_{MOS}$ | $3.23 \times 10^3$ | $8.33 \times 10^2$ | $2.67 \times 10^2$ | $8.02 \times 10^1$ | $1.33 \times 10^1$ | 5.77 | $1.45 \times 10^6$ | $5.84 \times 10^5$ |
| F2$_{MOS}$ | 6.00 | 3.46 | $1.03 \times 10^1$ | 2.31 | 0.00 | 0.00 | $6.40 \times 10^5$ | $1.31 \times 10^5$ |

Note(s): SD: Standard deviation; AREC: antibiotics resistant *E. coli*; ARFC: antibiotics resistant faecal coliform; ARVC: antibiotics resistant *Vibrio cholera*; ARTF: antibiotics resistant total heterotrophic flora.

### 3.4. Effect of Moringa oleifera Seeds on Antibiotic-Resistant Bacteria Abatement

Moringa oleifera seed powder was used as a natural coagulant in F0, F1 and F2 to evaluate its effect on bacteria load but also its effect on antibiotic-resistant bacteria. Results are presented in the table of data 3. Measured abatements after MO treatment in samples varied from one marker to another. The reduction in antibiotic-resistant bacteria in F0 was on average 91.7% for antibiotic-resistant *E. coli* (AREC), 92.2% for antibiotic-resistant faecal coliform (ARFC), 48.8% for antibiotic-resistant *Vibrio cholera* (ARVC) and 30.09% for antibiotic-resistant total heterotrophic flora (ARTF). In F1, reductions of 74.2%, 98%, 9.1% and 92% were obtained, respectively, for AREC, ARFC, ARVC and ARTF. The variable abatement rate was also noted in F2, with reduction of 97% for AREC, 93.9% for ARFC, 100% for ARVC and 6.3% for ARTF. Figure 4 shows the impact of MO's addition in wastewater on antibiotic-resistant bacteria removal. The effectiveness of Moringa treatment on ARB removal varied according to the bacterial species and the step of wastewater purification (ANOVA, $p < 0.01$). For AREC, MO's addition was the most effective in F2, with a 1.53 log reduction, whereas for ARFC and ARTF, the addition was most effective in F1, with a respective 1.68 and 1.09 log reduction. The effectiveness of MO's addition to ARVC was relatively low (0.04–0.29 log reduction), probably due to the low prevalence of *V. cholera* in the system. Based on these reductions and the abatement calculated between F0 & F1 and F1 & F2, projections were made and are given in Figure 5 with the aim of determining in which tank the addition of MO would be most effective for the improvement of the entire treatment process. Projections show that adding MO at the entrance of the infiltration well is more effective for AREC and ARTF, whereas for ARFC and ARVC, the best combination is the addition of MO in F1 (after decantation in the tank).

Additional observation was made on ESBL and carbapenem-resistant bacteria. After incubation, the abundances of ESBLEC and CR *E. coli* were $7 \pm 0$ and $1 \pm 0$ CFU Ml$^{-1}$ in F0$_{MOS}$; $7 \pm 0$ and $1 \pm 0$ CFU mL$^{-1}$ in F1$_{MOS}$ and not detected in F2$_{MOS}$. The abundance of ESBL and carbapenem resistance in the KESC group was $69 \pm 19$ and $440 \pm 28$ CFU mL$^{-1}$ in F0$_{MOS}$; $34 \pm 0$ and $7 \pm 0$ CFU mL$^{-1}$ in F1$_{MOS}$ and not detected in F2$_{MOS}$. The combined effect of the WWTP prototype with the use of MO as a natural coagulant in F2 helped to remove 100% of resistant KESC and *E. coli* in the WWTP prototype.

Additional studies on the effect of MO on antibiotic-resistant genes (ARG) need to be carried out, because the elimination of resistant bacteria does not necessarily include the elimination of ARG, which implies health risks if we consider a possible reuse of treated wastewater for irrigation. In a study conducted in 2013, Fahrenfeld et al. suggested that recycled water may be an important reservoir of ARGs, and ARG amplification is often detected during the distribution process when using wastewater for irrigation. They also

demonstrated that the chlorination step did not have any impact on ARG's elimination after wastewater treatment [38].

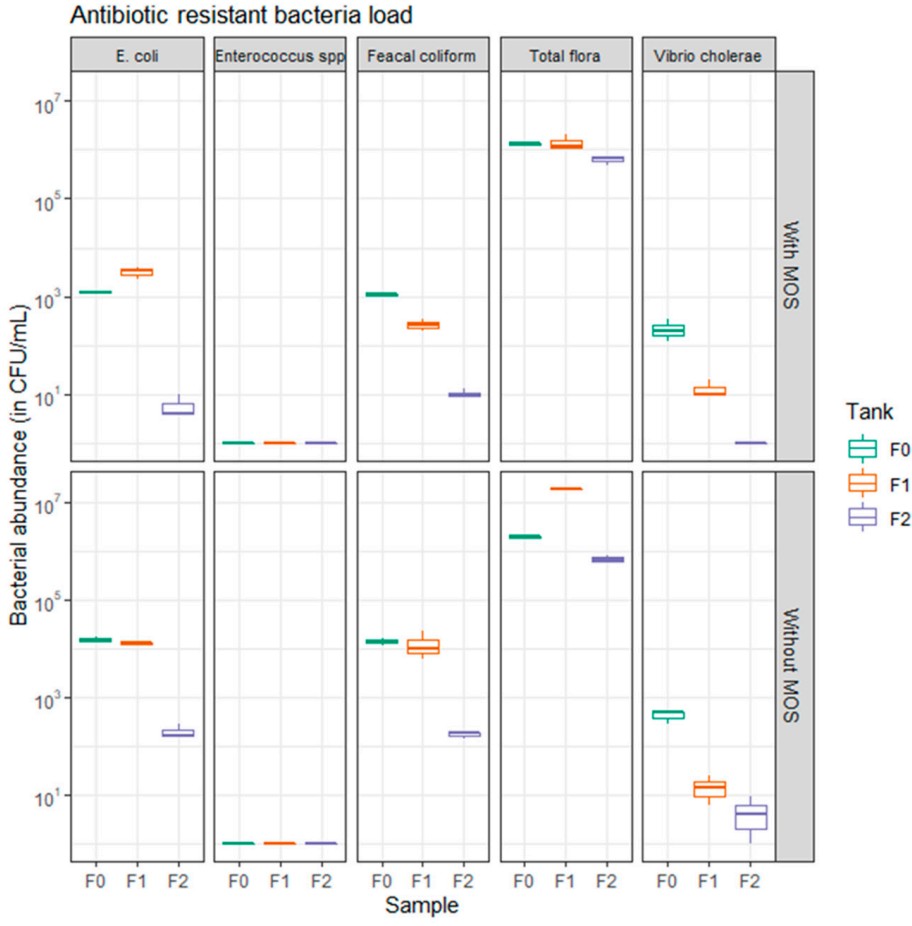

**Figure 4.** Abundances of antibiotic-resistant bacteria in samples taken at the entrance of each tank for evaluation of abatement; counted number of bacteria represented as log 10 reduction.

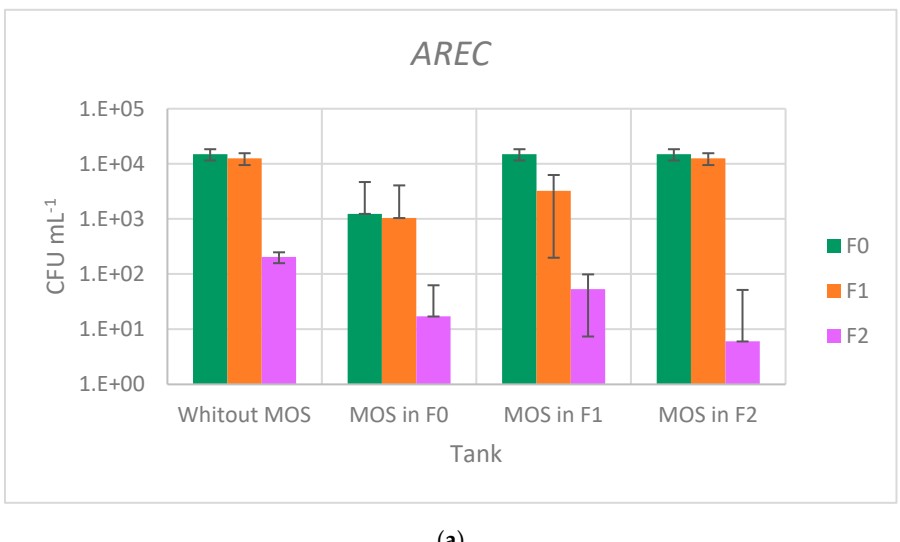

(**a**)

**Figure 5.** *Cont*.

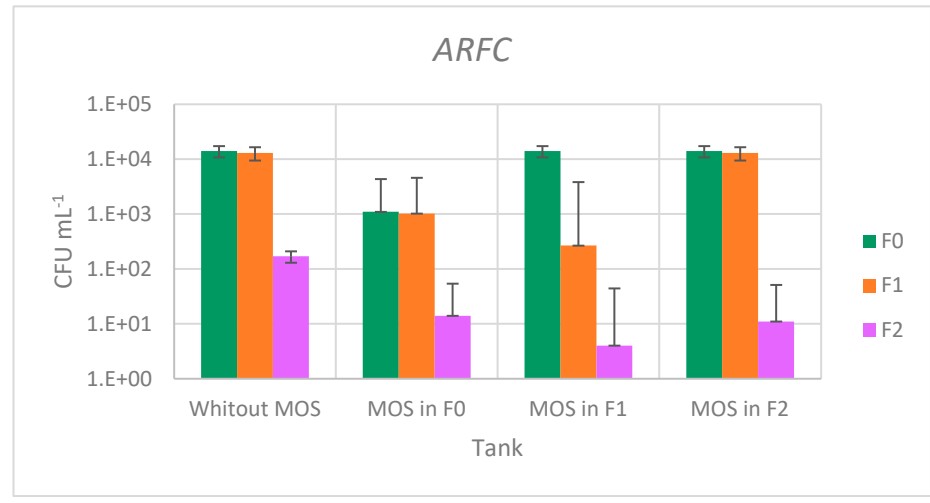

(**b**)

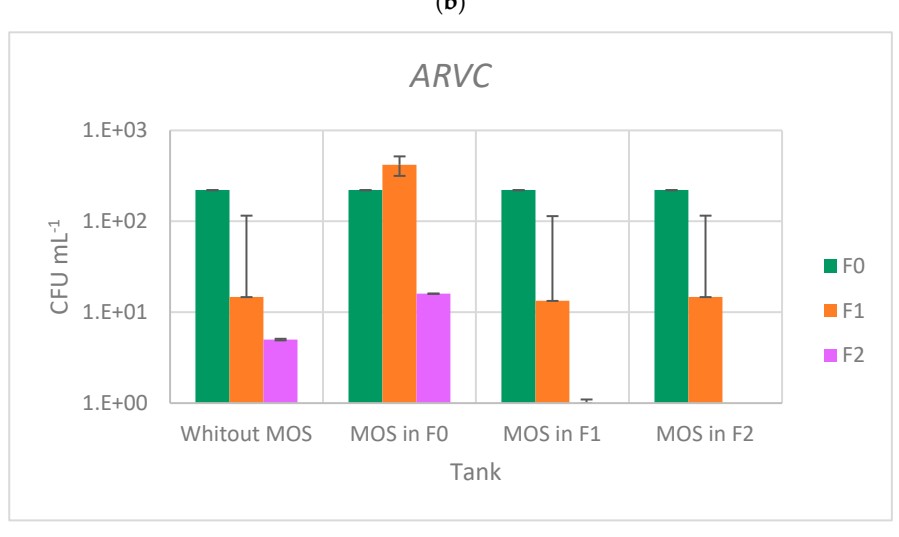

(**c**)

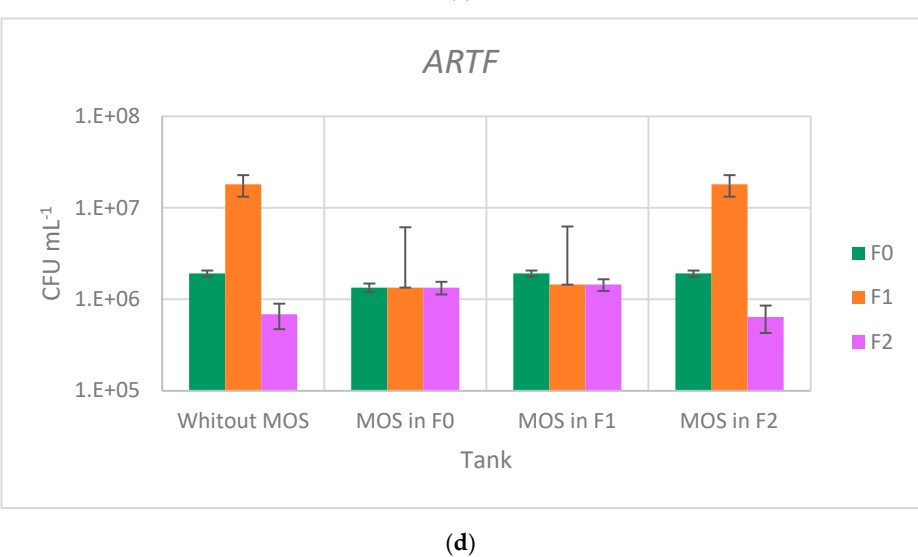

(**d**)

**Figure 5.** Projection of Moringa oleifera seeds' (MOs) effect on the entire treatment plant prototype after MOs' addition in F0, F1, F2; (**a**): abundance of antibiotic-resistant *E. coli* (AREC); (**b**): abundance of antibiotic-resistant faecal coliform (ARFC); (**c**): abundance of antibiotic-resistant Vibrio cholera (ARVC); (**d**): abundance of antibiotic-resistant total heterotrophic flora (ARTF).

## 4. Conclusions

The aim of this study was to evaluate the effectiveness of a well-managed wastewater treatment plant on reducing bacterial pollution and antibiotic-resistant bacteria, but also to assess the combined effect of the treatment plant with *Moringa oleifera* (MO) seeds used as a natural coagulant. After treatment with the WWTP prototype, the abundance of bacteria decreased about a percentage from 89.9% to 98.6% for FIB and up to 99.3% for *Vibrio cholera*, and removal from about 64.35% to 98.8% was noted for antibiotic-resistant FIB and up to 98.9% for antibiotic-resistant *Vibrio cholera*. Therefore, the counting number of *Enteroccocus* spp. and antibiotic-resistant *Enteroccocus* spp. were under the limit of quantification. The results indicate the strong efficiency of our WWTP prototype in reducing the microbiological load contained in domestic wastewaters. After the addition of MO into samples, the abatement of the bacteria load varied from about 28.3% to 92.4% for FIB and 35.3% to 95.6% for *Vibrio cholera*, and the percentages varied from 6.34% to 97.95% for antibiotic-resistant FIB and from 9.1% to 100% for antibiotic-resistant *Vibrio cholera*. The results highlight the positive impact of MO coagulation during domestic wastewater treatment processing. The efficiency of the WWTP prototype could be improved by an onsite addition of MO either at the entrance of the bacterial filter (F1) or at the entrance of the infiltration well (F2). It would also be important to identify the mechanisms that made the abatements such as flocculation observed by additional studies possible and elucidate insight into the components and chemical properties of the materials used in this study. The current work displays novelty in highlighting the impact of MO on antibiotic-resistant bacteria. Studies on the use of MOs as natural coagulants for wastewater treatment or on the proliferation of antibiotic-resistant bacteria are many, but to the best of our knowledge, this is the first study on the impact of MOs on antibiotic-resistant bacteria in a WWTP prototype under tropical conditions.

The authors' recommendations are to highlight the effectiveness of affordable unitary WWTP and its possible deployment for African developing countries and to display the benefits of using *Moringa oleifera* seeds as an alternative to chemical coagulants for wastewater treatment and the prevalence of antibiotic-resistant bacteria in WWTP.

**Author Contributions:** N.S., S.S., M.M., P.L.C. and J.P. conceived and designed the research; N.S., A.L., M.M. and J.P. performed the research (sampling and laboratory analysis); N.S., A.L., S.S., P.L.C. and J.P. analyzed the data. All authors wrote and have read, reviewed and approved the manuscript before submission. All authors have read and agreed to the published version of the manuscript.

**Funding:** Work by these authors was totally supported by the Schlumberger Foundation Faculty for the Future program. Ms. Sané was one of the 2020 laureates and obtained funding to carry out her PhD project for a sustainable solution for individual wastewater treatment in Africa.

**Data Availability Statement:** Not applicable.

**Acknowledgments:** We would also like to thank the University of Geneva for having made it possible to host Me Sané despite the health context and restrictions imposed in Europe. Further, we do not forget Campus France, which also allowed Me Sané to stay at the University of Bordeaux Montaigne. Thanks to The Bordeaux Imaging Center, who hosted me during my stay in France and trained me in the electronic microscopy method and sample preparation.

**Conflicts of Interest:** The authors declare no conflict of interest.

**Legal Entity:** The Schlumberger Stichting Fund—Parkstraat 83-89 2514JG The Hague the Netherlands. Registered—KvK Haaglanden No. 41167008 Tel.: +31-70-310-5400 Fax: +31-70310-5485.

**Appendix A**

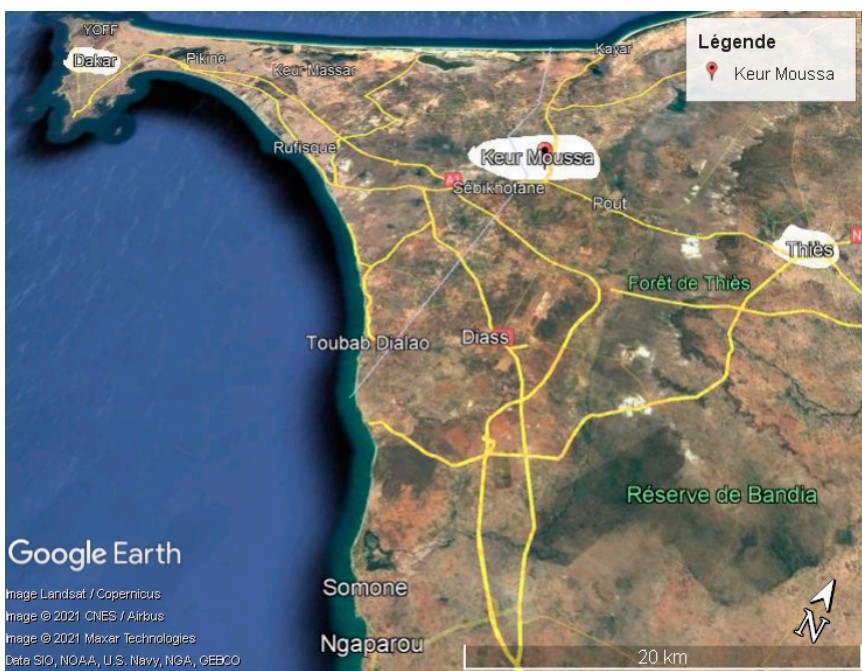

**Figure A1.** Google Map indicating the location of the rural community of Keur Moussa (studied site) in Senegal.

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
