# Peer review of "Development of a Process for Domestic Wastewater Treatment Using Moringa oleifera for Pathogens and Antibiotic-Resistant Bacteria Inhibition under Tropical Conditions"

_water, doi:10.3390/w14152379_

Round 1
Reviewer 1 Report
Reviewer Comments
Manuscript Number: Water-1780524
Authors of this manuscript studied the capacity of a wastewater treatment prototype system to eliminate faecal contamination and antibiotics resistant bacteria from domestic wastewater.
Authors have included and presented well some important results in the manuscript. However, the authors are requested to respond the following comments and suggestions to improve the quality of the manuscript further to publish in Water.
· Abstract: A well written abstract.
· Introduction; Page 2: In Line 79-80, authors stated “If we consider the French regulation [13], it sets the concentration of E. coli in reused wastewater depending on the sanitary quality targeted”. If the study based on the French regulations/standards, please rephrase the above statement appropriately
· Introduction; Page 2; Line 104: It was said that the study was carried out under tropical conditions in a sub-Saharan African country. In this case, what was the reason for complying with the French regulations. It is found that there are several studies have been carried out previously to study the impact of Moringa oleifera on the antibiotic resistant bacteria inhibition, but this could be the first study under “tropical conditions”
· Materials and Methods; Figure 1: Not very informative and not discussed in the text. Recommend submitting it as supplementary information
· Materials and Methods; Section 2.1: The basis of the design of the pilot system is not clear. Is it designed for the peak flow? Please specify the flowrate, hydraulic retention times and other design parameters considered to do the sizing the entry tank and the other reactors. It is also recommended to provide the relevant design standards or literature used to obtain these parameters in the manuscript
· Materials and Methods; Section 2.1: Please define the terms consistently. Is the entry tank same as the “All Purpose Tank” specified in the Figure 2(c)? Is this a fully closed tank and allows it to occur anaerobic biological treatment (like a conventional single compartment septic tank)?
· Materials and Methods; Section 2.3: Authors stated “the powder was immediately used for downstream applications. Subsequently, we put in contact wastewaters with MOS on the basis of a protocol preestablished by Abdallah LY in 2016”. What is meant by “downstream application”? What is the dosing rate? It is also noted that influent/effluent samples collected from the pilot system were investigated the impact of Moringa oleifera on the antibiotic resistant bacteria inhibition using a “batch study”. Experiment procedure is not very clearly stated. Please specify the experiments more clearly using variables, etc.
· Materials and Methods; Section 2.4: Microbiological analysis was limited to five species “Escherichia coli, Faecal coliforms, Enterococcus spp., Vibrio cholerae and Total Heterotrophic bacteria”. What was the basis of selection these species? How could the authors justify that this analysis is adequate, and the results are reliable?
· Results and discussion; Line 254-255: It was concluded stating that the highest abatements rates were obtained in samples F1 (at the entrance of the bacterial filter) for E. coli, Vibrio cholerae and total flora, while for faecal coliform it was obtained in F2. Please elaborate the possible reason/s for this finding. Is there a link with the reactor type for this finding?
· Results and discussion: Section 3.2 and the Figure 4 explains the projection of MOS's effect on the entire treatment plant prototype after MOS addition in F0, F1, F2; 4A. It looks the present results do not provide a strong conclusive finding. The results could impact on the dosing amount of MOS in each reactor, retention times of the reactors, composition of microorganisms and other operating conditions. Please specify this in the manuscript how these results could vary with and without tropical conditions
· Results and discussion; Section 3.4: Please specify the Table of Data discussed in this Section.
· Conclusions: Please specify a set of recommendations to perform future studies to obtain more reliable and accurate results to use for developing the technology further
· Conclusions: This section should be written to highlight the key findings from the present study clearly. It should not be included literature information (references) and a discussion. Please revise the conclusion section appropriately

Author Response
Dear Reviewer
please see the attachement of the fully revised version which includes corrections, modifications and english revision as suggested.
Best regards

Reviewer 2 Report
The manuscript on the topic Development of a process for domestic wastewater treatment using Moringa oleifera for pathogens and antibiotic resistant bacteria inhibition under tropical conditions is an interest to the reader and within the scope of the journal.
However, the manuscript needs revision prior to making any decision.
The comments are provided below for the author’s attention.
1. How the current work gain novelty needs to be highlighted in the script.
2. Motivation for using Moringa Oleifera seeds are not strongly addressed in the script. More elaborate discussion are needed related to the use of Moringa Oleifera seeds for water treatment.
3. Several typos and the grammatical mistake can be observed which need thorough correction.
4. Introduction section not convincing. It needs a more recent literature survey related to the current project.
5. On page 6 section 2.4. Microbial analysis of water samples, this section is not having a clear methodology described.
6. Any specific criteria for selection of Moringa Oleifera seeds use as a natural coagulant.
7. Is there any toxicity due to the use of Moringa Oleifera seeds
8. Section 3. Results and discussion, need to be more concise and to be clear.
9. Main findings need to be addressed and highlighted in the script.
Author Response
Dear reviewer
Please see attachment of the revised version of the manuscript which includes answers, corrections and modifications suggested.
Best regards
PL

Round 2
Reviewer 1 Report
I have briefly gone through the revised manuscript and I happy to accept it for publishing in Water now.Reviewer 2 Report
The authors have satisfactorily revised the manuscript.